# Allelopathic Toxicity of Cyanamide Could Control Amaranth (*Amaranthus retroflexus* L.) in Alfalfa (*Medicago sativa* L.) Field

**DOI:** 10.3390/molecules27217347

**Published:** 2022-10-28

**Authors:** Weihong Sun, Chaowei Yang, Xinhe Shan, Mingzhu An, Xianguo Wang

**Affiliations:** College of Grassland Science and Technology, China Agricultural University, Beijing 100193, China

**Keywords:** cyanamide, alfalfa, amaranth, allelopathic toxicity, oxidative damage, chlorophyll decrease

## Abstract

The inclination toward natural products has led to the onset of the discovery of new bioactive metabolites that could be targeted for specific therapeutic or agronomic applications. Despite increasing knowledge coming to light of allelochemicals as leads for new herbicides, relatively little is known about the mode of action of allelochemical-based herbicides on herbicide-resistant weeds. Cyanamide is an allelochemical produced by hairy vetch (*Vicia villosa* Roth.). This study aimed to detect the toxicity of cyanamide to alfalfa and amaranth. Seed germination experiments were carried out by the filter paper culture, and the seedling growth inhibition experiment was carried out by spraying alfalfa (*Medicago sativa* L.) and amaranth (*Amaranthus retroflexus* L.) seedlings with cyanamide. The results showed that when the concentration of cyanamide was 0.1 g·L^−1^, the germination of amaranth seeds could be completely inhibited without affecting the germination of alfalfa seeds. At the concentration of 0.5 g·L^−1^, cyanamide could significantly inhibit the growth of the root and stem of amaranth seedlings but did not affect the growth of alfalfa. This effect was associated with the induction of oxidative stress. The ascorbate peroxidase (APX) and catalase (CAT) activity of amaranth decreased by 6.828 U/g FW and 290.784 U/g FW, respectively. The malondialdehyde (MDA) content, peroxidase (POD), and superoxide dismutase (SOD) activity of amaranth firstly increased and then decreased with the increasing concentration of CA. These enzyme activities of amaranth changed more than that of alfalfa. Activities of the antioxidant enzymes APX, CAT, POD, and SOD and the content of MDA varied dramatically, thereby demonstrating the great influence of reactive oxygen species upon identified allelochemical exposure. In addition, cyanamide can also inhibit the production of chlorophyll, thereby affecting the growth of plants. From the above experiments, we know that cyanamide can inhibit the growth of amaranth in alfalfa fields. Thus, the changes caused by cyanamide described herein can contribute to a better understanding of the actions of allelochemical and the potential use of cyanamide in the production of bioherbicides.

## 1. Introduction

Alfalfa (*Medicago sativa* L.) is an important perennial forage crop with great economic value owing to its high biomass yield and quality forage, wide adaptability to different environments, and nitrogen fixation capacity [1]. However, alfalfa does not have a competitive advantage compared with weeds because of its slow growth in the seedling stage. Unfavorable weed control will reduce the density of alfalfa and even lead to failure of planting [2]. Alfalfa yields dropped 15 percent at 20 percent weed coverage and 59 percent at 40 percent weed coverage [3]. Amaranth (*Amaranthus retroflexus* L.) is the main weed in alfalfa fields and seriously harms the growth of alfalfa [4].

Amaranth is an annual herb, which originated from tropical America and has been widely distributed all over the world [5]. Amaranth with high phenotypic plasticity and genetic variability are adapted to live in various types of farmland as well as in weedy areas. The seeds of amaranth can form a persistent and stable seed bank because it grows rapidly and produces a large number of viable seeds [6]. Due to environmental and genetic factors, seeds have dormancy characteristics and uneven germination patterns, which can enhance adaptability and increase competitive advantage. Amaranth, which grows in alfalfa, wheat, corn, sugar beets, orchards, and vegetable gardens, can severely block sunlight and obstruct ventilation, depleting the soil of nutrients and thus inhibiting crop growth.

The application of chemical herbicides is a common way of controlling weeds, which has revolutionized weed control in the last hundred years, during which crop yields have grown significantly globally [7]. This inexpensive and effective method of weed control has become the primary method of clearing weeds in farmland, replacing manual, animal, and mechanical weed control. However, chemical herbicides are synthetic and non-natural compounds that are not easy to be degraded by microorganisms [8]. Their drawbacks are gradually revealed over time. The accumulation of chemical herbicides will cause environmental pollution and harm human health through the biological chain. At the same time, a large number of chemical pesticide residues penetrate into the ground, which can lead to groundwater contamination [9]. The more herbicides are used, the more resistant the weed becomes, so more herbicide needs to be used. It is a vicious circle.

For all the reasons listed above, it is necessary to develop natural herbicides with low toxicity and high efficiency that can be degraded. Allelochemicals can replace chemical herbicides to inhibit weed growth and reduce the use of chemical herbicides, which is conducive to the sustainable development of the environment [10].

Cyanamide (CA), also named hydrogen cyanamide (CN_2_H_2_), is an organic compound commonly applied in agriculture [11]. CA, as an allelochemical, was first found in hairy vetch (*Vicia villosa* Roth.), in all of its vegetative organs, including leaves, stems, roots, and in small amounts in seed endosperm [12]. It may be considered a source of inorganic nitrogen. Except for its fertilizing role, it is used as a pesticide, fungicide, or nematocide. The CA concentration in shoots was about 369–498 µg·g^−1^·FW, whereas in roots and endosperm was lower than 170 µg·g^−1^·FW. The study shows that phytotoxic CA can affect maize (*Zea mays* L.) root growth and root tip function [13]. CA was also applied for defoliation of cotton plants in order to facilitate harvesting of the bolls [14]. In this study, the allelopathic effect of CA was used to inhibit the growth of amaranth in an alfalfa field so as to achieve the effect of pollution-free and safe weed control.

## 2. Results and Discussion

### 2.1. Effects of CA on Seed Germination

In this study, we examined the effects of water solutions of CA on seed germination of alfalfa and amaranth. The inhibitory magnitude varied according to the concentration of CA. Seed germination of alfalfa and amaranth was not significantly affected when CA concentration was less than 0.01 g·L^−1^. CA has the effect of promoting growth at low concentrations and inhibiting growth at high concentrations on the germination of amaranth. Amaranth seemed to be more sensitive to CA than alfalfa, and the seeds imbibed in 0.1 g·L^−1^ CA had completely inhibited germination, and the response index was −100%, while the germination of alfalfa seeds was not affected, and response index was −6.43%. In contrast, when the CA concentration reached 0.4 g·L^−1^, the germination of alfalfa seeds was completely inhibited (Table 1, Figure 1). According to the non-linear fit of seed germination, the significant effect of CA on amaranth was markedly depicted at the concentration of 0.062 g·L^−1^ by inhibition of 50% compared to alfalfa (0.317%), and when the germination inhibition ratio of alfalfa was 50%, the concentration of CA was 0.313 g·L^−1^, which was five times that of amaranth (Figure 2A, Appendix A). In all cases, the hypocotyl length was more sensitive than radicle length. When the concentration of CA was 0.001 g·L^−1^, the inhibition ratio of the hypocotyl of amaranth and alfalfa was 57.09% and 5.06%, respectively, but the effect on radicle length was not significant. The hypocotyl length of amaranth was completely inhibited, and which inhibition ratio was 100% when the concentration of CA was 0.1 g·L^−1^ and the suppression rate of radicle growth was 93.10%. At the concentration of 0.001–0.05 g·L^−1^, although the growth of alfalfa radicle and hypocotyl was significantly inhibited compared with the control, there was no significant difference between the treatments. The growth reduction (GR50) of amaranth and alfalfa was 0.029 g·L^−1^ and 0.142 g·L^−1^, respectively, and the GR50 values of alfalfa was 4.9 times that of amaranth (Appendix A). With the increase of CA concentration, the fresh weight and dry weight of amaranth seed germination could be significantly inhibited, but there was no significant effect on alfalfa. When the CA concentration reached the highest 0.1 g·L^−1^, the synthetical allelopathic index (SE) showed that the inhibition ratio of the germination process of amaranth was as high as 98.27%, but the inhibition ratio of the germination process of alfalfa was only 12.39% (Table 1). The CA possesses the GR50 values for the seed germination process of amaranth, which was about 0.029 g·L^−1^. But at the same concentration of CA, the inhibition ratio of the alfalfa germination process was only 5.06%. And when the SE of alfalfa was −50%, the concentration of CA was 0.277 g·L^−1^, which was 9.55 times that of amaranth (Figure 2B, Appendix A).

Seed germination is often regarded as the starting point in the life cycle of plants, and it is also the most important and vulnerable stage [15]. The normal survival and reproduction of plants depend on the successful germination and growth of seeds into seedlings. Low concentrations of the *P. purpureum* aqueous extract (2%) and debris incorporated into the soil (25/500 g) inhibited the germination of the bioassay species (*E. indica*) [16]. Wu et al. showed that the inhibitory effects on seed germination of three test species (*Robinia pseudoacacia* L., *Lolium perenne* L., and *Lagerstroemia indica* L.) increased with increasing exposure times and at higher extract concentrations of *Mikania micrantha* [17]. Hussain et al. found that the *A. melanoxylon* flower extract (100%, 75%, and 50%) decreased the seed germination of *D. glomerata*, *R. acetosa*, *L. perenne*, and *L. sativa*. The flower extract caused the most reduction in the germination index and germination speed in *D. glomerata*, *L. perenne*, and *L. sativa* [18]. It could be explained by the entry of water through the integument during the germination process produced the entry of bioactive compounds by mass flow, which began their physiological activities to affect the germination and growth of seeds. The results of this experiment are similar to the above-mentioned studies. The seeds of amaranth showed a low germination rate, irregular germination, or even no germination under CA treatment. But at the same concentration, CA did not affect the germination of alfalfa seeds. Dorota et al. found that seed germination of the dicotyledonous plants was strongly inhibited by CA at both tested concentrations (1.2 mM, 3 mM) [19]. In this study, CA did not affect the germination of alfalfa, a dicotyledonous plant, which was inconsistent with the study of Dorota et al. This may be because the study by Dorota et al. did not include alfalfa, a dicotyledonous plant, whereas, in this study, we investigated the effect of CA on alfalfa seed germination in detail, indicating that some dicotyledonous plants generally were less sensitive to CA treatment. We conclude that CA has the potential for use as a natural herbicide to inhibit the germination of amaranth seeds in alfalfa fields.

### 2.2. Effect of CA on Seedling Growth

Different concentrations of CA (0 g·L^−1^, 0.1 g·L^−1^, 0.5 g·L^−1^, 0.75 g·L^−1^, 1 g·L^−1^, 1.25 g·L^−1^, 1.5 g·L^−1^) were applied at the two-leaf stage of alfalfa and four-leaf stage of amaranth. Obviously, compared with alfalfa, amaranth was more sensitive to CA. When the concentration of CA was 0.75–1.5 g·L^−1^, part of the amaranth gradually withered and died (Table 2, Figure 3). The root growth of amaranth was significantly inhibited when the concentration of CA was less than 0.75 g·L^−1^, while the root of alfalfa was not affected. The inhibition increased with increasing concentrations of CA (Table 2). The concentrations required for 50% growth reduction (IR50) for amaranth roots, as determined by the non-linear fit, were 1.383 g·L^−1^_,_ in contrast, the root growth inhibition ratio of alfalfa was only 21.539%, and the concentrations required for 50% growth reduction (IR50) for alfalfa roots was 1.757 g·L^−1^, which was 1.27 times that of amaranth (Figure 4A, Appendix A). When the concentration of CA was less than 1 g·L^−1^, there was no significant inhibitory effect on the growth of alfalfa stems, while the growth of amaranth stems was affected. With the increase in CA concentration, the inhibitory effect gradually strengthened. The inhibition ratio on the stem of amaranth was 51.5% at the highest concentration of 1.5 g·L^−1^ compared with 27.61% of alfalfa (Table 2). According to the non-linear fit of seedling growth, the significant effect of CA on amaranth stems was markedly depicted at the concentration of 1.432 g·L^−1^ by a 50% growth reduction (IR50) compared to alfalfa (21.271%), and the concentrations required for 50% growth reduction (IR50) for alfalfa stems was 1.769 g·L^−1^, which was 1.24 times that of amaranth (Figure 4B, Appendix A). At the concentration of 0.50–1.00 g·L^−1^, although the fresh weight and dry weight of alfalfa root and stem were not significantly inhibited compared with the control, with the increase of CA concentration, the fresh weight and dry weight of amaranth could be significantly inhibited (Table 2). When the CA concentration reached the highest 1.50 g·L^−1^, the SE showed that the inhibition rate of the seedling growth of amaranth was as high as 67.55%, but the inhibition rate of the seedling growth of alfalfa was only 39.18%. The CA possess the IR50 value for the seedling growth process of amaranth was about 0.915 g·L^−1^, but at the same concentration of CA, the inhibition rate of the alfalfa seedling growth process was only 11.75% (Figure 4C, Appendix A).

The plant often suffers from adverse environmental stresses during growth, such as salinity, drought, low temperature, etc. When plants were under stress, it was usually shown by their appearance, such as changing their morphological characteristics, slowing down their growth, and reconfiguring their biomass distribution. Treatment of onion roots with CA resulted in a decrease in root growth rate accompanied by a decrease in the accumulation of fresh and dry weight. In CA-treated onion, the following changes were detected: reduction of mitotic cells, inhibition of proliferation of meristematic cells and cell cycle, and modifications of cytoskeleton arrangement [20]. Treatment of tomato (*Solanum lycopersicum* L.) roots with CA (1.2 mM) resulted in inhibition of growth accompanied by alterations in cell division and imbalance of plant hormone (ethylene and auxin) homeostasis. Moreover, the phytotoxic effect of CA was also manifested by modifications in expansin gene expression, especially in expansins responsible for cell wall remodeling after the cytokinesis (LeEXPA9, LeEXPA18) [21]. In this study, when the concentration of CA was 1.5 g·L^−1^, the suppression rate of root and stem growth of amaranth was 61.1% and 51.5%, respectively, and there was a drastic check on the fresh weight and dry weight, and even led to the death of some amaranth. Allelochemicals may also affect plant growth at the cellular level. Considering the effects of CA on the growth of onion and maize, the reason why CA inhibited the growth of amaranth may be that CA inhibited the cell division of amaranth, resulting in its slow growth. The restriction of plant growth in response to allelopathic compounds may have different bases, e.g., the inhibition of enzymatic activity, damage of cellular membranes, a disturbance in ion uptake and water balance, etc. [22]. Modifications of plant growth and development in response to exposure to allelochemicals may reflect alterations in the molecular biology of cells, their ultrastructure, or their biochemical and physiological processes.

### 2.3. Effects of CA on Activation of Antioxidant Defensive Enzymes

Plants respond to oxidative stress via the rapid stimulation of enzymatic antioxidant defense components. In this study, the antioxidant enzyme activities, including ascorbate peroxidase (APX), catalase (CAT), malondialdehyde (MDA), peroxidase (POD), and superoxide dismutase (SOD), were more responsive in amaranth seedlings under CA compared to alfalfa seedlings (Figure 5).

The APX activity of amaranth decreased with the increase in CA concentration. Compared with the control, the activity of APX decreased by 6.828 U/g FW from 8.262 U/g FW without treatments to 1.434 U/g FW. The APX activity of alfalfa climbed up and then declined, and then increased, increased from 5.527 U/g FW to 5.535 U/g FW with the increase of CA concentration, which indicated that the APX activity gets slowly back on track (Figure 5A).

The CAT activity of amaranth gradually decreased with the increase of CA concentration and fell from 307.083 U/g FW to 16.299 U/g FW, which indicated that the activity of CAT was severely damaged, and most of the CAT enzymes of amaranth were almost inactivated. The CAT activity of alfalfa increased gradually with the increase of CA concentration, rising from 149.09 U/g FW to 456.527 U/g FW. This indicated that the key enzyme CAT of the alfalfa biodefense system began to respond to oxidative stress (Figure 5B).

The MDA content of amaranth firstly increased and then decreased with the increasing concentration of CA. The MDA content reached 59.086 nmoL·g^−1^ FW when the CA concentration was 1.25 g·L^−1^, and when the CA concentration was 1.5 g·L^−1^, the MDA content was 41.171 nmoL·g^−1^ FW. This is probably because free radicals in amaranth acted on lipids and underwent peroxidation resulting in excessive MDA, with the CA acting on amaranth. When the CA concentration was up to 1.5 g·L^−1^, the damage to the plant membrane was more serious, and some even died. Therefore, when the CA concentration was the highest, the content of MDA decreased. The MDA content of alfalfa also increased first and then decreased with the increase of CA concentration, but only increased from 24.939 nmoL·g^−1^ FW to 31.696 nmoL·g^−1^ FW and then decreased to 25.568 nmoL·g^−1^ FW, which indicated that CA had little effect on the membranous of alfalfa (Figure 5C).

With the increase of CA concentration, the POD activity of amaranth first increased and then decreased, from 242.306 U/g FW in the control group to 1733.361 U/g FW, and finally decreased to 45.565 U/g FW, which changed obviously. POD is an important physiological indicator of plant tissue aging, which indicated that CA accelerated the senescence of amaranth and began to wither and die with the increase of CA concentration. The POD of alfalfa increased by only 175.097 U/g FW, from 9295.056 U/g FW to 9470.153 U/g FW, with the increase of CA concentration. This showed that CA had little effect on the POD of alfalfa (Figure 5D).

With the increase of CA concentration, the SOD activity of amaranth first increased and then decreased, from 183.117 U/g FW to 352.116 U/g FW, and then decreased to 152.776 U/g FW, while that of alfalfa decreased from 175.558 U/g FW to 97.678 U/g FW. The SOD activity of amaranth changed more than that of alfalfa, which indicated that amaranth was more seriously affected by CA (Figure 5E).

Plants will produce a lot of reactive oxygen species (ROS) under adversity, which will cause tissue damage through several different cellular and molecular pathways. ROS also acts as a signal molecule of chemical defense, inducing plants to produce signal cascades and activating the antioxidant system of the body for defense [23]. The excessive production of antioxidant enzymes in treated plants under allelochemical stress has evolved amaranth and alfalfa, a complex system of enzymatic antioxidants in order to reduce the induced oxidative. Among them, APX, CAT, POD, and SOD are important reactive oxygen species scavenging enzymes in the antioxidant system. Under the catalysis of SOD, the superoxide anion radical generates H_2_O_2_ through a disproportionation reaction [24]. CAT and POD can catalyze H_2_O_2_ to form H_2_O and O_2_. These three aspects function together to remove excess free radicals in the body, thereby improving the stress resistance of plants [25]. APX is a key enzyme for scavenging H_2_O_2_, and it is essential for protecting chloroplasts and other cellular components from damage by H_2_O_2_ and the hydroxyl radicals it produces [26]. MDA is one of the commonly used indicators to measure the degree of oxidative stress. It can reflect the degree of plant membrane lipid peroxidation [27]. In an organism, free radicals act on lipids to undergo peroxidation, and the final product of oxidation is MDA. This will cause cross-linking polymerization of life macromolecules such as proteins and nucleic acids, and its production is toxic and can aggravate cell membrane damage. From our study, amaranth improved the antioxidant activity by increasing the enzymes compared to alfalfa. Stimulation of antioxidant activity is commonly associated with enhanced stress tolerance in plants.

Some identified allelochemicals activated the CAT activity in maize seedlings and cucumber cotyledons [28]. In this study, CA activated the activity of CAT in alfalfa. With the increase in CA concentration, CAT formed a defense system to protect cells from being poisoned by H_2_O_2_. However, CA did not activate the CAT activity of the amaranth, which may be because, with the increase of CA concentration, the amaranth was seriously damaged, so the CAT enzyme activity was damaged or even inactivated. In addition, the enhanced activity of SOD of amaranth indicated that excessive generation of O^2−^ had been triggered by CA, which was then upregulated to mitigate oxidative damage. MDA is one of the commonly used indicators to measure the degree of oxidative stress. The MDA activity of amaranth was higher than that of alfalfa, which indicated that amaranth had suffered more serious oxidative stress damage. POD is an important respiratory enzyme in plants. Measuring peroxidase activity can help understand the evolutionary stage and evolutionary trend of plants. When the concentration of CA was 1.5 g·L^−1^, the POD activity of alfalfa was 208 times higher than that of amaranth, which indicated that alfalfa had stronger resistance to CA.

### 2.4. Effects on Photosynthetic Activities

The mean in vivo chlorophyll-a fluorescence of alfalfa and amaranth was measured after exposure to different concentrations of CA. There was no apparent change in amaranth compared to the control group when the concentration of CA was less than 0.5 g·L^−1^. With the increase of CA concentration, the content of chlorophyll-a in amaranth gradually decreased. The chlorophyll-a content decreased by 62.86% after seven days of exposure to the high concentration of CA. When the concentration of CA was less than 1.00 g·L^−1^, it had no significant effect on chlorophyll-a of alfalfa. When the CA concentration was the highest at 1.50 g·L^−1^, the content of alfalfa chlorophyll-a decreased by 42.05% (Figure 6A). When the CA concentration was 0.5 g·L^−1^, the chlorophyll-b of amaranth was not affected compared with the control group. With the increase of CA concentration, the content of amaranth chlorophyll-b gradually decreased. When the CA concentration reached the highest at 1.5 g·L^−1^, the content of chlorophyll B decreased to 68.52% of the control group. However, when the CA concentration was lower than 1 g·L^−1^, it did not affect the content of alfalfa chlorophyll-b, and when the CA concentration reached the highest 1.5 g·L^−1^, it decreased to 40.40% compared with the control group (Figure 6B). When the concentration of CA is 1 g·L^−1^, it had no effect on the carotenoid content of alfalfa but significantly affected the content of amaranth carotenoid. The carotenoids of alfalfa and amaranth were decreased by 37.42% and 54.50%, respectively, compared with the control group when the concentration of CA was 1.5 g·L^−1^ (Figure 6C).

Photosynthesis is an important part of material transfer and energy flow in an ecosystem. The level of chlorophyll content is an important indicator to reflect the strength of plant photosynthesis. Chlorophyll can absorb energy from light and convert carbon dioxide into carbohydrates [29]. Usually, 90–95% of the output of plants comes from photosynthesis. It can be said that photosynthesis plays a decisive role in plant growth. Therefore, we can judge the growth status of plants by measuring the content of chlorophyll. The total chlorophyll content of *N. wightii* was significantly reduced in all plants treated with both aqueous seed and leaf extracts of *D. stramonium* [30]. Sunflower has a strong allelopathic effect on *Sinapis arvensis* and *Setaria viridis*, which can lead to a decrease in chlorophyll content during growth [31]. The results were consistent with their conclusion that the contents of chlorophyll-A, chlorophyll-B, and carotene of alfalfa and amaranth were significantly decreased under the treatment of CA. This may be due to the enhancement of chlorophyllase activity caused by CA, thus accelerating the degradation of chlorophyll.

Either CA inhibits the activity of chlorophyll-related enzymes or reduces its intermediates. CA can also cause oxidative stress and generate a large number of oxygen free radicals, resulting in the destruction of the molecular structure of chlorophyll, thereby reducing the content of chlorophyll.

## 3. Materials and Methods

### 3.1. Experimental Materials

CA (purity 95%) for the experiment of seed germination and seedlings growth was purchased from Beijing Bailingwei Technology Co., Ltd. Amaranth seeds used in this experiment were collected in Hulunbuir, Inner Mongolia, on the 6 October 2021. The variety of alfalfa is WL168, which was purchased from Beijing Zhengdao Seed Industry Co., Ltd. (Beijing, China).

### 3.2. Seeds Germination Assays

Seeds of alfalfa and amaranth were sterilized with 0.2% sodium hypochlorite solution for 10 min and rinsed with distilled water three times to avoid pathogen contamination [32]. Fifty seeds from the above-tested species were equidistantly placed in Petri dishes (diameter = 90 mm) with two layers of filter paper and then treated with 10 mL of the eight CA at different concentrations (0 g·L^−1^, 0.001 g·L^−1^,0.005 g·L^−1^,0.01 g·L^−1^,0.05 g·L^−1^,0.1 g·L^−1^,0.2 g·L^−1^,0.4 g·L^−1^) independently. Distilled water was used as the control, and three replicates of each treatment group were established. The dishes were then cultured in a universal environmental test chamber at a constant temperature of 25 ± 0.5 °C, 85% humidity, and a controlled 12 h light/12 h dark cycle. Sealing the petri dish with parafilm to maintain the humidity of the filter paper in the petri dish. The number of germinated seeds was counted on the seventh day. In addition, the root length, stem length, and biomass of each treatment were also measured.

### 3.3. Pot Experiment

For soil preparation, the ratio of nutrient soil, vermiculite, and perlite were 4:1:1. We poured 800 g of soil into plastic bowls with an upper mouth diameter of 11.5 cm, lower mouth diameter of 8 cm, and height of 11 cm. Then an equal amount of water was added into each treatment group to soak the soil and seeds were sown 4 h later. Sixty seeds of amaranth and thirty seeds of alfalfa were sown independently in each bowl with three biological repeats. After the emergence of alfalfa and amaranth, they were thinned, and 15 seedlings were kept in each pot. Different concentrations of CA (0 g·L^−1^, 0.1 g·L^−1^, 0.5 g·L^−1^, 0.75 g·L^−1^, 1 g·L^−1^, 1.25 g·L^−1^, 1.5 g·L^−1^) were applied at the two-leaf stage of alfalfa and four-leaf stage of amaranth. Plant heights, biomass, enzyme activity, and chlorophyll of alfalfa and amaranth were measured at seven days of spraying CA.

### 3.4. Antioxidant Enzyme Activities

Fresh leaves of alfalfa and amaranth (200 mg) (from each treatment or control) were homogenized in 10 mL of sodium phosphate buffer (0.1 M, pH 7.0) to extract antioxidant enzymes. The homogenized material was centrifuged at 13,000 rpm for 30 min at 4 °C, and the supernatant was collected for the enzymatic analyses of superoxide dismutase (SOD), catalase (CAT), and ascorbate peroxidase (APX), peroxidase (POD) and the content of malondialdehyde (MDA), which was quantified from spectrophotometer readings. The final volume of the reaction for reading enzyme activity was 2 mL [33]. All readings were conducted in triplicate.

Superoxide dismutase was assayed according to the following method proposed by Giannopolitis and Ries [34]. The absorbance was determined at 560 nm, and one unit of SOD activity was defined as the quantity of the enzyme that inhibits nitro-blue tetrazolium (NBT) photoreduction by 50%. CAT activity was measured as per the method of Cakmak and Marschner [35]. The reaction mixture (2 mL) consisted of 25 mM phosphate buffer (pH 7.0), 10 mM H_2_O_2_ and 0.2 mL of enzyme extract. The activity was determined by measuring the rate of disappearance of H_2_O_2_ for 1 min at 240 nm and calculated using an extinction coefficient of 39.4 mM^−1^·cm^−1^ and expressed as enzyme units U/g·FW. One unit of CAT activity was defined as the amount of enzyme that catalyzes the decomposition of 1 µM min^−1^ of H_2_O_2_.

### 3.5. Concentrations of Chl-a, Chl-b, Carotenoids

The content of Chl-a, Chl-b, and carotenoids was determined according to Hung et al. [36]. Fresh leaves of alfalfa and amaranth that have been frozen in liquid nitrogen were ground with a small amount of basic magnesium carbonate powder and 90% acetone. The mixture was poured into a centrifuge tube of 10 mL and centrifuged at 4000× *g* for 10 min at 4 °C. Afterwards, the supernatant was removed into a volumetric flask of 10 mL. The grinding process was repeated 2–3 times. All the supernatants were gathered into the same volumetric flask, where acetone was added to make the total volume of 10 mL. Finally, the absorbance was measured at 440, 644, and 662 nm with a spectrophotometer (722, China) and acetone which were used as a blank solution. The concentrations of Chl-a, Chl-b, and carotenoids were calculated according to the following equations:Ca = (9.784 × D_662_ − 0.990 × D_644_) × V/m 
Cb = (21.426 × D_644_ − 4.650 × D_662_) × V/m 
Cc = [(4.695 × D_440_ − 0.268 × (Ca + Cb)] × V/m
where V is the volume of the extract (mL), and m is the quality of fresh leaves.

### 3.6. Statistical Analysis

Each treatment set was composed of three replicate tests. The test data were expressed as the mean ± standard deviation. The analysis and calculation of experimental data were processed using SPSS 22.0. Experimental data were subjected to one-way analysis of variance (ANOVA) and post hoc LSD tests to determine significant differences among mean values at the probability level of 0.05. And according to the relevant data to make the relevant trend chart of seed germination and seedling growth.

## 4. Conclusions

Cyanamide was responsible for the phytotoxic effect, promoting a drastic reduction in radicle and hypocotyl lengths, as well as the frequency of germinated seeds of amaranth. The allelochemical action was mainly evident from the lowest concentration of 0.1 g·L^−1^. The germination of amaranth seeds could be completely inhibited without affecting the germination of alfalfa seeds. At the concentration of 0.5 g·L^−1^, cyanamide could significantly inhibit the growth of the root and stem of amaranth seedlings but did not affect the growth of alfalfa. This effect was associated with the induction of oxidative stress. Activities of the antioxidant enzymes APX, CAT, POD, SOD, and the content of MDA varied dramatically, thereby demonstrating the great influence of reactive oxygen species upon identified allelochemical exposure. In addition, cyanamide can also inhibit the production of chlorophyll, thereby affecting the growth of plants. This study comprehensively expounded that the allelopathic toxicity of cyanamide could control amaranth in alfalfa fields from two aspects of seed germination and seedling growth. This provides a theoretical basis for the development of cyanamide as a new bioherbicide.

## Figures and Tables

**Figure 1 molecules-27-07347-f001:**
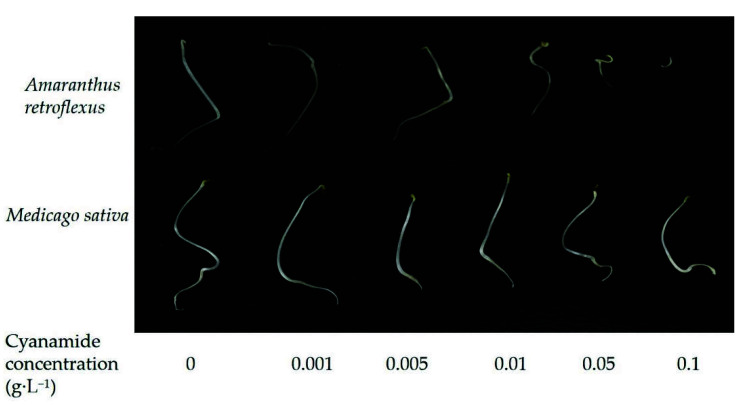
The effects of alfalfa and amaranth seeds treated with different concentrations of cyanamide for 7 days.

**Figure 2 molecules-27-07347-f002:**
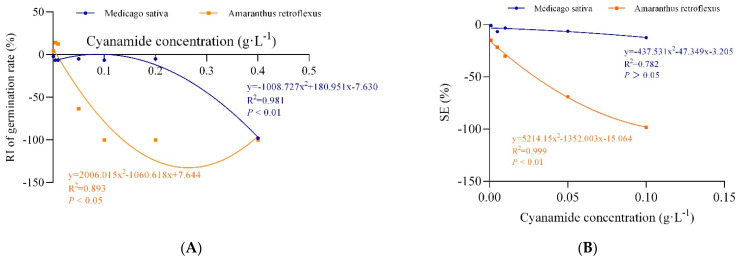
The non-linear fit of response index of germination rate (RI of germination rate) and the synthetical allelopathic index (SE). (**A**) The non-linear fit of response index of germination rate (RI of germination rate). (**B**) The non-linear fit of synthetical allelopathic index (SE).

**Figure 3 molecules-27-07347-f003:**
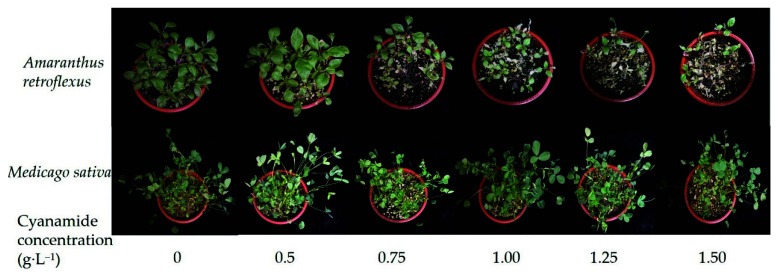
The effects of the two-leaf stage of alfalfa and the four-leaf stage of amaranth applied by different concentrations of CA for seven days.

**Figure 4 molecules-27-07347-f004:**
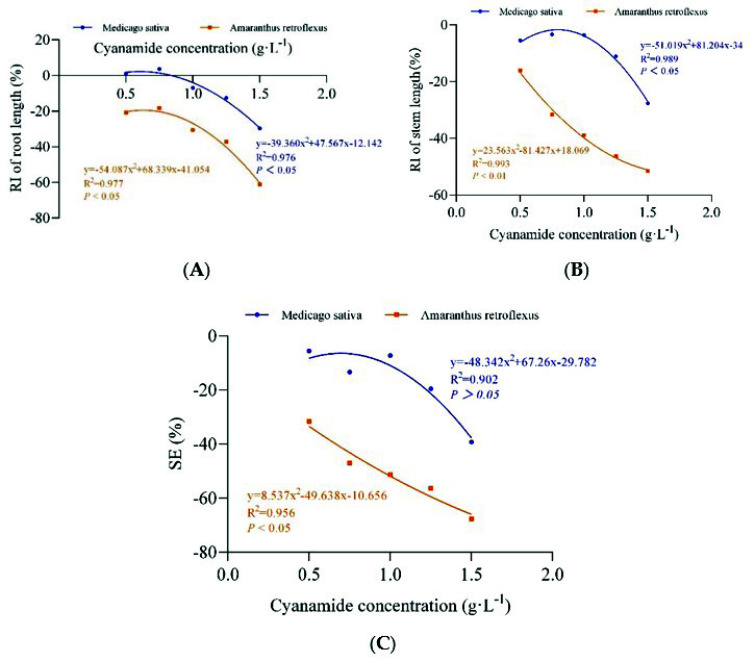
The non-linear fit of the response index of seedling root length (RI of root length), stem length (RI of stem length), and the synthetical allelopathic index (SE). (**A**) The non-linear fit of response index of root length (RI of root length). (**B**) The non-linear fit of response index of (RI of stem length). (**C**) The non-linear fit of synthetical allelopathic index (SE).

**Figure 5 molecules-27-07347-f005:**
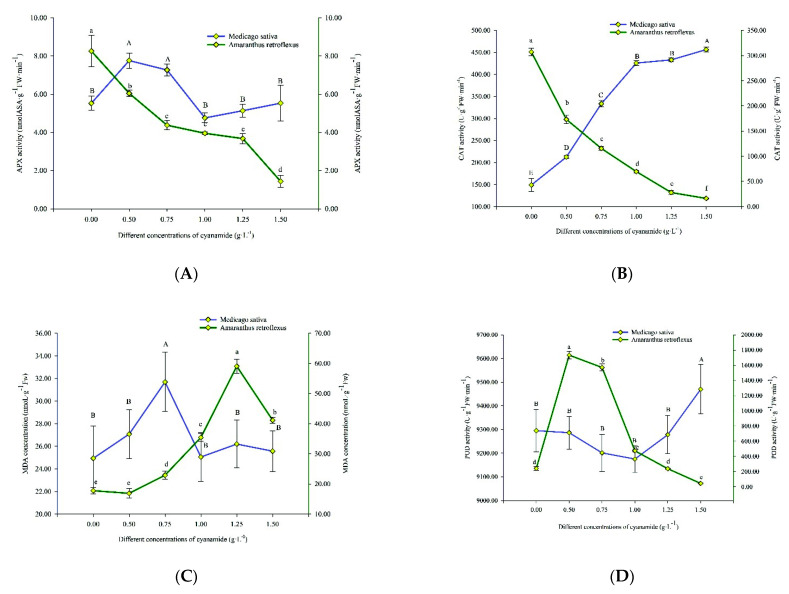
Ascorbate peroxidase (APX), catalase (CAT), peroxidase (POD), superoxide dismutase (SOD) activities, and malondialdehyde (MDA) content in alfalfa and amaranth exposed to CA. (**A**) The ascorbate peroxidase (APX) activity in alfalfa and amaranth exposed to CA. (**B**) The catalase (CAT) activity in alfalfa and amaranth exposed to CA. (**C**) The peroxidase (POD) activity in alfalfa and amaranth exposed to CA. (**D**) The superoxide dismutase (SOD) activity in alfalfa and amaranth exposed to CA. (**E**) The malondialdehyde (MDA) content in alfalfa and amaranth exposed to CA. Different letters in columns indicate significant differences among treatments at *p* < 0.05 (LSD test).

**Figure 6 molecules-27-07347-f006:**
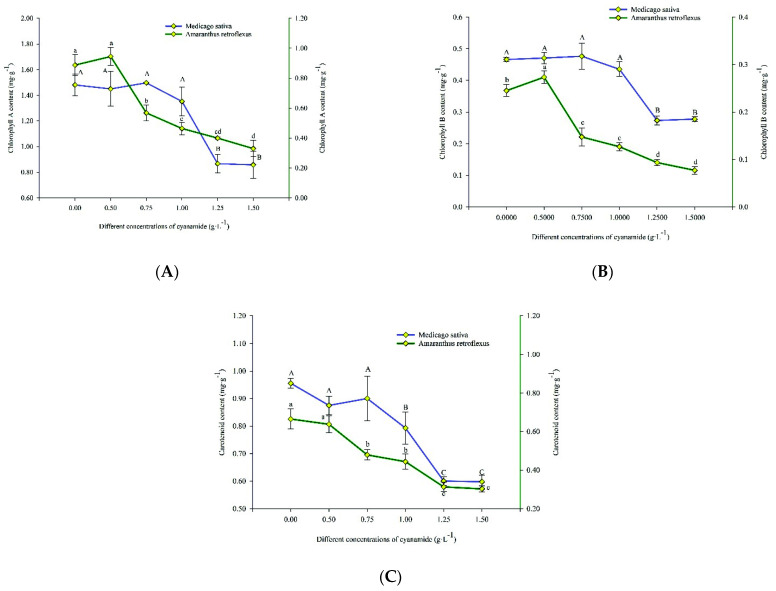
The content of chlorophyll A, chlorophyll B and carotenoids in alfalfa and amaranth exposed to CA. (**A**) The content of chlorophyll A in alfalfa and amaranth exposed to CA. (**B**) The content of chlorophyll B in alfalfa and amaranth exposed to CA. (**C**) The content of carotenoids in alfalfa and amaranth exposed to CA. Different letters in columns indicate significant differences among treatments at *p* < 0.05 (LSD test).

**Table 1 molecules-27-07347-t001:** Phytotoxic effect of cyanamide on seed germination of alfalfa and amaranth at different concentration (0, 0.001, 0.005, 0.01, 0.05, and 0.1 g·L^−1^). Different lower-case letters in columns indicate significant differences among treatments of amaranth at *p* < 0.05 (LSD test). Different capital letters in columns indicate significant differences among treatments of alfalfa at *p* < 0.05 (LSD test).

	Cyanamide Concentration (g·L^−1^)	Germination Rate (%)	RI (%)	Radicle Length (cm)	RI (%)	Hypocotyl Length (cm)	RI (%)	Fresh Weight (g)	RI (%)	Dry Weight (g)	RI (%)	SE (%)
*Amaranthus retroflexus*	0	47.33 ± 4.16 a		1.45 ± 0.14 a		2.01 ± 0.36 a		0.0033 ± 0.00098 a		0.0004 ± 0.00015 a		
	0.001	49.33 ± 9.02 a	4.23	1.61 ± 0.17 a	11.16	0.86 ± 0.08 b	−57.09	0.0027 ± 0.0002 ab	−18.18	0.0001 ± 0.00001 ab	−75.00	−14.97
	0.005	54.00 ± 7.21 a	14.08	1.50 ± 0.23 a	3.80	0.65 ± 0.11c	−67.85	0.0021 ± 0.00006 bc	−36.36	0.0001 ± 0.00002 ab	−75.00	−21.58
	0.01	53.33 ± 10.07 a	12.68	0.96 ± 0.16 b	−33.72	0.56 ± 0.11 c	−72.25	0.0024 ± 0.0001 b	−27.27	0.0002 ± 0.00008 b	−50.00	−30.14
	0.05	17.33 ± 6.11 b	−63.38	0.39 ± 0.08 c	−72.84	0.36 ± 0.05 d	−82.02	0.0014 ± 0.00046 c	−57.58	0.0001 ± 0.00001 c	−75.00	−68.96
	0.1	0.00 ± 0.00 c	−100	0.1 ± 0.09 d	−93.10	0.00 ± 0.00 e	−100	0.00 ± 0.00 d	−100	0.00 ± 0.00 c	−100.00	−98.27
*Medicago sativa*	0	93.33 ± 1.15 A		1.55 ± 0.12 A		4.55 ± 0.18 A		0.0245 ± 0.0015 A		0.0014 ± 0.00010 A		
	0.001	91.33 ± 3.06 A	−2.14	1.52 ± 0.13 AB	−1.94	4.32 ± 0.12 B	−5.06	0.0257 ± 0.0016 A	4.90	0.0014 ± 0.00021 A	0.00	−0.85
	0.005	87.33 ± 4.16 A	−6.43	1.35 ± 0.11 BC	−12.73	3.51 ± 0.19 C	−19.09	0.0257 ± 0.0012 A	4.90	0.0014 ± 0.00025 A	0.00	−6.67
	0.01	87.33 ± 3.06 A	−6.43	1.53 ± 0.21 AB	−0.76	4.22 ± 0.13 B	−7.29	0.0242 ± 0.0021 A	−1.22	0.0014 ± 0.00035 A	0.00	−3.14
	0.05	88.67 ± 2.31 A	−5.00	1.36 ± 0.10 ABC	−12.19	4.30 ± 0.19 B	−5.42	0.0240 ± 0.0018 A	−2.04	0.0013 ± 0.00015 A	−7.14	−6.36
	0.1	87.33 ± 4.16 A	−6.43	1,32 ± 0.18 C	−14.56	3.72 ± 0.28 C	−18.28	0.02.7 ± 0.0014 B	−15.51	0.0013 ± 0.00026 A	−7.14	−12.39

**Table 2 molecules-27-07347-t002:** Phytotoxic effect of cyanamide on seedling growth of alfalfa and amaranth at different concentration (0, 0.50, 0.75, 1.00, 1.25, and 1.50 g·L^−1^). Different lower-case letters in columns indicate significant differences among treatments of amaranth at *p* < 0.05 (LSD test). Different capital letters in columns indicate significant differences among treatments of alfalfa at *p* < 0.05 (LSD test).

	Cyanamide Concentration (g·L^−1^)	Root Length (cm)	RI (%)	Stem Length (cm)	RI (%)	Fresh Weight (g)	RI (%)	Dry Weight (g)	RI (%)	SE (%)
*Amaranthus retroflexus*	0	4.018 ± 0.518 a		7.533 ± 0.345 a		0197 ± 0.010 a		0.067 ± 0.002 a		
	0.50	3.183 ± 0.249 b	−20.78	6.328 ± 0.201 b	−16.00	0.101 ± 0.007 b	−48.40	0.040 ± 0.006 b	−40.90	−31.52
	0.75	3.285 ± 0.530 b	−18.25	5.155 ± 0.224 c	−31.57	0.051 ± 0.007 c	−73.89	0.024 ± 0.003 c	−63.88	−46.90
	1.00	2.793 ± 0.630 bc	−30.49	4.605 ± 0.257 d	−38.87	0.058 ± 0.004 c	−70.48	0.023 ± 0.002 c	−65.07	−51.23
	1.25	2.523 ± 0.303 c	−37.20	4.045 ± 0.240 e	−46.31	0.059 ± 0.009 c	−70.13	0.019 ± 0.002 d	−71.34	−56.25
	1.50	1.563 ± 0.391 d	−61.10	3.653 ± 0.377 f	−51.50	0.034 ± 0.007 d	−82.95	0.017 ± 0.002 d	−74.63	−67.55
*Medicago sativa*	0	10.850 ± 0.718 A		12.720 ± 0.760 A		0.430 ± 0.036 A		0.054 ± 0.003 A		
	0.50	10.973 ± 0.436 A	1.14	12.028 ± 0.766 AB	−5.44	0.350 ± 0.024 A	−18.53	0.055 ± 0.005 A	0.92	−5.90
	0.75	11.252 ± 0.468 A	3.70	12.298 ± 0.804 AB	−3.32	0.221 ± 0.018 A	−48.58	0.052 ± 0.004 A	−4.96	−13.29
	1.00	10.105 ± 0.324 B	−6.87	12.270 ± 0.730 AB	−3.54	0.376 ± 0.033 A	−12.63	0.051 ± 0.003 A	−5.51	−7.14
	1.25	9.485 ± 0.603 B	−12.58	11.313 ± 0.937 B	−11.06	0.311 ± 0.027 B	−27.70	0.040 ± 0.003 B	−26.47	−19.45
	1.50	7.632 ± 0.539 C	−29.66	9.208 ± 0.779 C	−27.61	0.210 ± 0.010 C	−51.28	0.028 ± 0.000 C	−48.16	−39.18

## Data Availability

The data presented in this study are available in Appendix A.

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
