# Peer review of "Allelopathic Toxicity of Cyanamide Could Control Amaranth (Amaranthus retroflexus L.) in Alfalfa (Medicago sativa L.) Field"

_molecules, 2022, doi:10.3390/molecules27217347_

Round 1

Reviewer 1 Report

Presented me for review paper entitled: Allelopathic toxicity of cyanamide could control amaranth (Amaranthus retroflexus L.) in alfalfa (Medicago sativa L.) field has the following issues. 

First, the abstract is very unclear in presentingthe story behind it.  The following sentence: Activities of the antioxidant enzymes APX, CAT, MDA, POD, and SOD varied dramatically, thereby demonstrating the great influence of reactive oxygen species upon identified allelochemical exposure.  

Why exactly is the set of enzymes listed in the abstract a great influence? without any data? The abstract should be rephrased. Very vague and unclear. 

In the Intro part, cire the following: Kozuharova, Ekaterina, et al. "MULTIVARIATE STATISTICAL CLASSIFICATION OF PLANT FEATURES--THE CASE WITH ONOBRYCHIS PINDICOLA SUBSP. URUMOVII DEGEN & DREN." Comptes rendus de l'Académie bulgare des Sciences, vol. 70, no. 11, Nov. 2017, AND

https://doi.org/10.1515/chem-2019-0082

https://doi.org/10.7546/CRABS.2022.07.04

Extend the Inro comment about the statistical methods of such a study. 

The logistic regression analysis of response index of seed germination (RI) and the syn-thetical allelopathic index (SE).  -  very low quality, not clear explanatin about! 

Explane what was the main criteria for assesing the effec of the following enzimes: APX, CAT, MDA, POD and SOD, ? 

WHat was the main difference between. 

Explain more in the details the sentance, and very poor english: 

Reactive oxygen species (ROS) will increase in plants under adverse conditions, which will not only be toxic to plant cells, but also act as a signal molecule for chemical defense, inducing plants to produce a signal cascade reaction and initiating the body's antioxidant system for defense [23]. 

Statistical Analysis 

Each treatment set was composed of three parallel tests -  explain why? 

Nothing was shown in the SI as a used sets. 

The table with the basic statistics should be provide. 

Author Response

Response to Reviewer 1 Comments

Manuscript ID: molecules-1971042

Title: Allelopathic toxicity of cyanamide could control amaranth (Amaranthus retroflexus L.) in alfalfa (Medicago sativa L.) field 

Dear Reviewer,

  Thank you for giving us the opportunity to submit a revised draft of the manuscript “Allelopathic toxicity of cyanamide could control amaranth (Amaranthus retroflexus L.) in alfalfa (Medicago sativa L.) field” for publication in the Molecules. We have carefully studied the comments and revised the manuscript accordingly (the revised text is highlighted by track changes). Please find below point to point responses to the comments and suggestions of the reviewer.

We look forward to hearing from you regarding our submission. We would be glad to respond to any further questions and comments that you may have.

Sincerely,

                                                                                                  Dr. Weihong Sun

Please find the following response to the comments and suggestions of the reviewer:

  1. First, the abstract is very unclear in presentingthe story behind it. The following sentence: Activities of the antioxidant enzymes APX, CAT, MDA, POD, and SOD varied dramatically, thereby demonstrating the great influence of reactive oxygen species upon identified allelochemical exposure.  

Why exactly is the set of enzymes listed in the abstract a great influence? without any data? The abstract should be rephrased. Very vague and unclear. 

Response: We have added data on related enzyme activities to the abstract. “ The ascorbate peroxidase (APX) and catalase (CAT) activity of amaranth decreased by 6.828 U/g FW and 290.784 U/g FW, respectively. And the  malondialdehyde (MDA) content, peroxidase (POD) and superoxide dismutase (SOD) activity of amaranth firstly increased and then decreased with the increasing of the concentration of CA. These enzymes activities of amaranth changed more than that of alfalfa. Activities of the antioxidant enzymes APX, CAT, POD, SOD and content of MDA varied dramatically, thereby demonstrating the great influence of reactive oxygen species upon identified allelochemical exposure.”

  1. In the Intro part, cire the following: Kozuharova, Ekaterina, et al. "MULTIVARIATE STATISTICAL CLASSIFICATION OF PLANT FEATURES--THE CASE WITH ONOBRYCHIS PINDICOLA SUBSP. URUMOVII DEGEN & DREN." Comptes rendus de l'Académie bulgare des Sciences, vol. 70, no. 11, Nov. 2017, AND

https://doi.org/10.1515/chem-2019-0082

https://doi.org/10.7546/CRABS.2022.07.04

Extend the Inro comment about the statistical methods of such a study. 

Response:We carefully read these three literatures, which involve the application of cluster analysis and principal component analysis (PCA). However, We did not use these methods to analyze the data, so I do not know which part of the introduction is appropriate for these articles. Please give me your detailed guidance.

  1. The logistic regression analysis of response index of seed germination (RI) and the syn-thetical allelopathic index (SE).  -  very low quality, not clear explanation about! 

Response: We have made the figures about logistic regression analysis again and modified it in the article.

  1. Explane what was the main criteria for assessing the effectof the following enzimes: APX, CAT, MDA, POD and SOD?  What was the main difference between. 

Response: Plants will produce a lot of reactive oxygen species (ROS) under adversity, which will cause tissue damage through several different cellular and molecular pathways. ROS also acts as a signal molecules of chemical defense, inducing plants to produce signal cascades and activating the antioxidant system of body for defense. The excessive production of antioxidant enzymes in treated plant under allelochemicals stress have evolved amaranth and alfalfa a complex system of enzymatic antioxidant in order to reduce the induced oxidative. Among them, APX (Ascorbate peroxidase), CAT (Catalase), POD (Peroxidase) and SOD (Superoxide dismutase) are important reactive oxygen species scavenging enzymes in the antioxidant system. Under the catalysis of SOD, superoxide anion radical generates H2O2 through disproportionation reaction. CAT and POD can catalyze H2O2 to form H2O and O2. These three aspects function together to remove excess free radicals in the body, thereby improving the stress resistance of plants. APX is a key enzyme for scavenging H2O2,and it is essential for protecting chloroplasts and other cellular components from damage by H2O2 and the hydroxyl radicals it produces. MDA is one of the commonly used indicators to measure the degree of oxidative stress, it can reflect the degree of plant membrane lipid peroxidation. In an organism, free radicals act on lipids to undergo peroxidation, and the final product of oxidation is malondialdehyde. This will cause cross-linking polymerization of life macromolecules such as proteins and nucleic acids, and its production is toxic and can aggravate cell membrane damage.

The functions and effects of APX, CAT, MD, POD and SOD have been explained in detail in the discussion.

  1. Explain more in the details the sentance, and very poor english: 

Plants will produce a lot of reactive oxygen species (ROS) under adversity, which will cause tissue damage through several different cellular and molecular pathways. ROS also acts as a signal molecules of chemical defense, inducing plants to produce signal cascades and activating the antioxidant system of body for defense. 

Response: Plants will produce a lot of reactive oxygen species (ROS) under adversity, which will cause tissue damage through several different cellular and molecular pathways. ROS also acts as a signal molecules of chemical defense, inducing plants to produce signal cascades and activating the antioxidant system of body for defense.

6. Statistical Analysis: Each treatment set was composed of three parallel tests -  explain why? 

Response: We apologize for clarity of statement, and we have changed “three parallel tests” to “three replicate tests”.

7. Nothing was shown in the SI as a used sets. The table with the basic statistics should be provide. 

Response: Table 1 and Table 2 are the data about the effect of cyanamide on seed germination and seedling growth of alfalfa and amaranth, mainly including germination rate, radicle length, hypocotyl length, root length, stem length, fresh weight, dry weight, etc. These are the most basic data of plant growth. Although these data are basic, they are very important, and for the sake of the integrity of the article, we do not think they should be included in the SI.

In addition, we have made some amendments to the content of the article.

  • We have added data on related enzyme activities to the abstract. “ The ascorbate peroxidase (APX) and catalase (CAT) activity of amaranth decreased by 6.828 U/g FW and 290.784 U/g FW, respectively. And the  malondialdehyde (MDA) content, peroxidase (POD) and superoxide dismutase (SOD) activity of amaranth firstly increased and then decreased with the increasing of the concentration of CA. These enzymes activities of amaranth changed more than that of alfalfa.Activities of the antioxidant enzymes APX, CAT, POD, SOD and content of MDA varied dramatically, thereby demonstrating the great influence of reactive oxygen species upon identified allelochemical exposure.”(page 1)
  • We have changed 'Amaranth' ·by 'amaranth' in the article.(page 1)
  • We have changed ' allelopathic toxicity'by ' Allelopathic toxicity' in the article. (page 1)
  • The revised sentence is“The seeds of amaranth can form a persistent and stable seed bank because it grows rapidly and produces a large number of viable seeds”. (page 1)
  • We have changed 'is' ·by 'was' in the article.(page 2)
  • The revised sentence is“Amaranth seemed to be more sensitive to CA than alfalfa” . (page2)
  • The revised sentence is“When the concentration of CA was 0.001 g·L-1, the ibhibition rates of hypocotyl of amaranth and alfalfa were 57.09% and 5.06%, respectively, but the effect on radicle length was not significant.” (page2-3)
  • We have changed 'Amaranth' ·by 'amaranth' in the article.(page 4-5)
  • We have changed 'But' ·by 'but' in the article.(page 5)
  • The revised sentence is“When plants was under stress, it was usually shown by its appearance, such as changing its morphological characteristics, slowing down its growth and reconfiguring its biomass distribution.” (page 5)
  • The revised sentence is“Plants respond to oxidative stress via the rapid stimulation of enzymatic antioxidant defense components. In this study, the antioxidant enzyme activities, including ascorbate peroxidase (APX), catalase (CAT), malondialdehyde (MDA), peroxidase (POD) and superoxide dismutase (SOD) were more responsive in amaranth seedlings under CA compared to alfalfa seedlings (Figure 5). ” (page 7)
  • We changed the unit of CAT/SOD/APX/POD to U/g FW, and changed the unit ofMDA to nmol/g FW. (page 7)
  • We have changed 'O2−' ·by 'O2' in the article. (page 8)
  • The revised sentence is“The MDA activity of amaranth was higher than that of alfalfa, which indicated that amaranth had suffered more serious oxidative stress damage.” (page 8)
  • The revised sentence is“Ascorbate peroxidase (APX), catalase (CAT), peroxidase (POD), superoxide dismutase (SOD) activities and malondialdehyde (MDA) content in alfalfa and amaranth exposed to CA. ” (page 9)
  • We have changed 'the concentrationof chlorophyll-a' ·by 'the content of chlorophyll-a' in the article. (page 9)
  • We have changed 'carotene' ·by 'carotenoid' in the article.(page 9)
  • The revised sentence is“The homogenized material was centrifuged at 13,000 rpm for 30 min at 4℃, and the supernatant was collected for the enzymatic analyses of superoxide dismutase (SOD), catalase (CAT), and ascorbate peroxidase (APX), peroxidase (POD) and the content of malondialdehyde (MDA), which was quantified from spectrophotometer readings.” (page 11)
  • We have changed 'Catalase' ·by 'CAT' in the article.(page 11)
  • We have changed 'grinded' ·by 'ground' in the article.(page 11)
  • The revised sentence is“Activities of the antioxidant enzymes APX, CAT, , POD,SOD and content of MDA varied dramatically, thereby demonstrating the great influence of reactive oxygen species upon identified allelochemical exposure.” (page 12)
  • The revised sentence is“Weihong Sun and Xinhe Shan contributed to write the original draft.” (page 12)
  • We have changed 'conflilict' ·by 'conflict' in the article.(page 12)
  • We have also modified the format of some references, which will not be listed in detail.(page 12-14)

Reviewer 2 Report

I have carefully read the manuscript entitled ‘Allelopathic toxicity of cyanamide could control amaranth (Amaranthus retroflexus L.) in alfalfa (Medicago sativa L.) fieldl’. The introduction is well written and the results are convincing. In my opinion, the present version of the manuscript needs to be slightly revised before possible publication in the ‘molecules’ journal, as reported below.

-change ‘ug·g-1’·by ‘µg·g-1’,

- add the full name of the antioxidant enzyme (see paragraph 2.3.)

- use the correct symbols to indicate units of measure (umolASA ?)

- increases the resolution of the figures

Author Response

Response to Reviewer 2 Comments

Manuscript ID: molecules-1971042

Title: Allelopathic toxicity of cyanamide could control amaranth (Amaranthus retroflexus L.) in alfalfa (Medicago sativa L.) field 

Dear Reviewer,

  Thank you for giving us the opportunity to submit a revised draft of the manuscript “Allelopathic toxicity of cyanamide could control amaranth (Amaranthus retroflexus L.) in alfalfa (Medicago sativa L.) field” for publication in the Molecules. We have carefully studied the comments and revised the manuscript accordingly (the revised text is highlighted by track changes). Please find below point to point responses to the comments and suggestions of the reviewer.

We look forward to hearing from you regarding our submission. We would be glad to respond to any further questions and comments that you may have.

Sincerely,

Dr. Weihong Sun

Please find the following response to the comments and suggestions of the reviewer:

  1. change ‘ug·g-1’·by ‘µg·g-1

Response: We have changed 'ug·g-1' ·by 'µg·g-1' in the article.

  1. add the full name of the antioxidant enzyme (see paragraph 2.3.)

Response: We have added the full name of the antioxidant enzyme.

  1. use the correct symbols to indicate units of measure (umolASA ?)

Response: We are very sorry for the wrong use of the unit. Now we have changed ‘umolASA’ by ‘U/g FW’.

  1. increases the resolution of the figures.

Response: Thank you very much for your advice, and we have  increased the resolution of the figures.

In addition, we have made some amendments to the content of the article.

  • We have added data on related enzyme activities to the abstract. “ The ascorbate peroxidase (APX) and catalase (CAT) activity of amaranth decreased by 6.828 U/g FW and 290.784 U/g FW, respectively. And the  malondialdehyde (MDA) content, peroxidase (POD) and superoxide dismutase (SOD) activity of amaranth firstly increased and then decreased with the increasing of the concentration of CA. These enzymes activities of amaranth changed more than that of alfalfa.Activities of the antioxidant enzymes APX, CAT, POD, SOD and content of MDA varied dramatically, thereby demonstrating the great influence of reactive oxygen species upon identified allelochemical exposure.” (page 1)
  • We have changed 'Amaranth' ·by 'amaranth' in the article.(page 1)
  • We have changed ' allelopathic toxicity'by ' Allelopathic toxicity' in the article. (page 1)
  • The revised sentence is“The seeds of amaranth can form a persistent and stable seed bank because it grows rapidly and produces a large number of viable seeds”. (page 1)
  • We have changed 'is' ·by 'was' in the article.(page 2)
  • The revised sentence is“Amaranth seemed to be more sensitive to CA than alfalfa” . (page2)
  • The revised sentence is“When the concentration of CA was 0.001 g·L-1, the ibhibition rates of hypocotyl of amaranth and alfalfa were 57.09% and 5.06%, respectively, but the effect on radicle length was not significant.” (page2-3)
  • We have changed 'Amaranth' ·by 'amaranth' in the article.(page 4-5)
  • We have changed 'But' ·by 'but' in the article.(page 5)
  • The revised sentence is“When plants was under stress, it was usually shown by its appearance, such as changing its morphological characteristics, slowing down its growth and reconfiguring its biomass distribution.” (page 5)
  • The revised sentence is“Plants respond to oxidative stress via the rapid stimulation of enzymatic antioxidant defense components. In this study, the antioxidant enzyme activities, including ascorbate peroxidase (APX), catalase (CAT), malondialdehyde (MDA), peroxidase (POD) and superoxide dismutase (SOD) were more responsive in amaranth seedlings under CA compared to alfalfa seedlings (Figure 5). ” (page 7)
  • We changed the unit of CAT/SOD/APX/POD to U/g FW, and changed the unit ofMDA to nmol/g FW. (page 7)
  • We have changed 'O2−' ·by 'O2' in the article. (page 8)
  • The revised sentence is“The MDA activity of amaranth was higher than that of alfalfa, which indicated that amaranth had suffered more serious oxidative stress damage.” (page 8)
  • The revised sentence is“Ascorbate peroxidase (APX), catalase (CAT), peroxidase (POD), superoxide dismutase (SOD) activities and malondialdehyde (MDA) content in alfalfa and amaranth exposed to CA. ” (page 9)
  • We have changed 'the concentrationof chlorophyll-a' ·by 'the content of chlorophyll-a' in the article. (page 9)
  • We have changed 'carotene' ·by 'carotenoid' in the article.(page 9)
  • The revised sentence is“The homogenized material was centrifuged at 13,000 rpm for 30 min at 4℃, and the supernatant was collected for the enzymatic analyses of superoxide dismutase (SOD), catalase (CAT), and ascorbate peroxidase (APX), peroxidase (POD) and the content of malondialdehyde (MDA), which was quantified from spectrophotometer readings.” (page 11)
  • We have changed 'Catalase' ·by 'CAT' in the article.(page 11)
  • We have changed 'grinded' ·by 'ground' in the article.(page 11)
  • The revised sentence is“Activities of the antioxidant enzymes APX, CAT, , POD,SOD and content of MDA varied dramatically, thereby demonstrating the great influence of reactive oxygen species upon identified allelochemical exposure.” (page 12)
  • The revised sentence is“Weihong Sun and Xinhe Shan contributed to write the original draft.” (page 12)
  • We have changed 'conflilict' ·by 'conflict' in the article.(page 12)
  • We have also modified the format of some references, which will not be listed in detail.(page 12-14)

Reviewer 3 Report

This article by Wang and co-workers describes a study to determine the possible toxicity of cyanamide to two important plants in agriculture, alfalfa and amaranth. The authors applied different concentrations of cyanamide over both plants, and detected an interesting selective inhibition of germination and growth of amaranth without effects over alfalfa. This fact showed that cyanamide is a good inhibitor of amaranth growth. Additionally, we the authors used antioxidant enzymes, the results completely changed, showing an important effect of oxygen in the process.   

Generally, this is good work that provides important information about cyanamide effects in the inhibition of germination and growth of weeds, that it is crucial from an agriculture perspective. Thus, I recommend accepting this article under MINOR revision.

Some considerations must be taken into account.

  1. On page 4, Figure 2, it is very difficult to see Charts A and B. The size is too small, and the quality is too low. The authors must improve this figure.
  2. On page 6, Figure 4, the same problems appeared. The authors must also improve this figure.
  3. Many typing mistakes can be detected in the text. A full revision is necessary.

Author Response

Response to Reviewer 3 Comments

Manuscript ID: molecules-1971042

Title: Allelopathic toxicity of cyanamide could control amaranth (Amaranthus retroflexus L.) in alfalfa (Medicago sativa L.) field 

Dear Reviewer,

  Thank you for giving us the opportunity to submit a revised draft of the manuscript “Allelopathic toxicity of cyanamide could control amaranth (Amaranthus retroflexus L.) in alfalfa (Medicago sativa L.) field” for publication in the Molecules. We have carefully studied the comments and revised the manuscript accordingly (the revised text is highlighted by track changes). Please find below point to point responses to the comments and suggestions of the reviewer.

We look forward to hearing from you regarding our submission. We would be glad to respond to any further questions and comments that you may have.

Sincerely,

Dr. Weihong Sun

Please find the following response to the comments and suggestions of the reviewer:

  1. On page 4, Figure 2, it is very difficult to see Charts A and B. The size is too small, and the quality is too low. The authors must improve this figure.

On page 6, Figure 4, the same problems appeared. The authors must also improve this figure.

Response: Thank you very much for your advice, and we have  increased the resolution of the figures.

  1. Many typing mistakes can be detected in the text. A full revision is necessary.

We have corrected many mistakes in the article.

  • We have added data on related enzyme activities to the abstract. “ The ascorbate peroxidase (APX) and catalase (CAT) activity of amaranth decreased by 6.828 U/g FW and 290.784 U/g FW, respectively. And the  malondialdehyde (MDA) content, peroxidase (POD) and superoxide dismutase (SOD) activity of amaranth firstly increased and then decreased with the increasing of the concentration of CA. These enzymes activities of amaranth changed more than that of alfalfa.Activities of the antioxidant enzymes APX, CAT, POD, SOD and content of MDA varied dramatically, thereby demonstrating the great influence of reactive oxygen species upon identified allelochemical exposure.”(page 1)
  • We have changed 'Amaranth' ·by 'amaranth' in the article.(page 1)
  • We have changed ' allelopathic toxicity'by 'Allelopathic toxicity' in the article. (page 1)
  • The revised sentence is“The seeds of amaranth can form a persistent and stable seed bank because it grows rapidly and produces a large number of viable seeds”. (page 1)
  • We have changed 'is' ·by 'was' in the article.(page 2)
  • The revised sentence is“Amaranth seemed to be more sensitive to CA than alfalfa” . (page 2)
  • The revised sentence is“When the concentration of CA was 0.001 g·L-1, the ibhibition rates of hypocotyl of amaranth and alfalfa were 57.09% and 5.06%, respectively, but the effect on radicle length was not significant.” (page 2-3)
  • We have changed 'Amaranth' ·by 'amaranth' in the article.(page 4-5)
  • We have changed 'But' ·by 'but' in the article.(page 5)
  • The revised sentence is“When plants was under stress, it was usually shown by its appearance, such as changing its morphological characteristics, slowing down its growth and reconfiguring its biomass distribution.” (page 5)
  • The revised sentence is“Plants respond to oxidative stress via the rapid stimulation of enzymatic antioxidant defense components. In this study, the antioxidant enzyme activities, including ascorbate peroxidase (APX), catalase (CAT), malondialdehyde (MDA), peroxidase (POD) and superoxide dismutase (SOD) were more responsive in amaranth seedlings under CA compared to alfalfa seedlings (Figure 5). ” (page 7)
  • We changed the unit of CAT/SOD/APX/POD to U/g FW, and changed the unit ofMDA to nmol/g FW. (page 7)
  • We have changed 'O2−' ·by 'O2' in the article. (page 8)
  • The revised sentence is“The MDA activity of amaranth was higher than that of alfalfa, which indicated that amaranth had suffered more serious oxidative stress damage.” (page 8)
  • The revised sentence is“Ascorbate peroxidase (APX), catalase (CAT), peroxidase (POD), superoxide dismutase (SOD) activities and malondialdehyde (MDA) content in alfalfa and amaranth exposed to CA. ” (page 9)
  • We have changed 'the concentrationof chlorophyll-a' ·by 'the content of chlorophyll-a' in the article. (page 9)
  • We have changed 'carotene' ·by 'carotenoid' in the article.(page 9)
  • The revised sentence is“The homogenized material was centrifuged at 13,000 rpm for 30 min at 4℃, and the supernatant was collected for the enzymatic analyses of superoxide dismutase (SOD), catalase (CAT), and ascorbate peroxidase (APX), peroxidase (POD) and the content of malondialdehyde (MDA), which was quantified from spectrophotometer readings.” (page 11)
  • We have changed 'Catalase' ·by 'CAT' in the article.(page 11)
  • We have changed 'grinded' ·by 'ground' in the article.(page 11)
  • The revised sentence is“Activities of the antioxidant enzymes APX, CAT, , POD,SOD and content of MDA varied dramatically, thereby demonstrating the great influence of reactive oxygen species upon identified allelochemical exposure.” (page 12)
  • The revised sentence is“Weihong Sun and Xinhe Shan contributed to write the original draft.” (page 12)
  • We have changed 'conflilict' ·by 'conflict' in the article.(page 12)
  • We have also modified the format of some references, which will not be listed in detail.(page 12-14)

Round 2

Reviewer 1 Report

The article was slightly improved, but still, Figures 2, 3, and 4 are very unclear. Present in a better way plot or split the plots and present the part of that in the SI.

About the Intro, the idea is also to comment on other methods. Also, I think that only based on a regression excluding the classification was a very weak point of the design of the paper.

Still, don't thin that the paper is ready to be published at all.

Author Response

  1. The article was slightly improved, but still, Figures 2, 3, and 4 are very unclear. Present in a better way plot or split the plots and present the part of that in the SI.

Response: Thank you very much for your advice, and we have redone the non-linear fit to the data and increased the resolution of the figures. Apart from that, we performed the non-linear fit of cyanamide on seed germination and seedling growth of alfalfa and amaranth. Data of the seed germination process included germination rate, radicle length, hypocotyl length, fresh weight, dry weight and the synthetical allelopathic index (SE). Data of the seedling growth process included root length, stem length, fresh weight, dry weight and the synthetical allelopathic index (SE). Finally, based on the non-linear fit, the concentration of cyanamide with inhibition rate of 50% in the process of germination and growth was calculated, which can better compare the allelopathic toxicity of cyanamide to alfalfa and amaranth. We made charts by non-linear fit of cyanamide with each index as supporting information, which made the content of the article more substantial and convincing.

The non-linear fit refers to the functional relationship between coordinates expressed by discrete points on the plane approximated by continuous curves. We used the non-linear fit method to investigate the toxic effect of cyanamide concentration on alfalfa and amaranth, and calculated the concentration of cyanamide when the inhibitory rate of cyanamide toxicity on alfalfa and amaranth was 50% according to the obtained formula, so as to compare the toxicity of cyanamide to alfalfa and amaranth.

At the same time, we have made corresponding modifications to the content of the article.

  • According to the non-linear fit of seed germination, the significant effect of CA on amaranth was markedly depicted at the concentration of 0.062 g·L-1 by an inhibition of 50% compared to alfalfa (0.317%), and when the germination inhibition ratio of alfalfa was 50%, the concentration of CA was 0.313 g·L-1, which was 5 times that of amaranth (Figure 2A, Table 3). (page 2-3)
  • The growth reduction (GR50) of amaranth and alfalfa was 0.029 g·L-1 and 0.142 g·L-1, respectively, and the GR50 values of alfalfa was 4.9 times that of amaranth (Table 3). (page 3)
  • The CA possess the GR50 values for seed germination process of amaranth was about 0.029 g·L-1. But at the same concentration of CA, the inhibition ratio of alfalfa germination process was only 5.06%. And when the SE of alfalfa was -50%, the concentration of CA was 0.277 g·L-1, which was 9.55 times that of amaranth (Figure 2B, Table 3). (page 3)
  • The concentrations required for 50% growth reduction (IR50) for amaranth roots, as determined by the non-linear fit, were 1.383 g·L-1, in contrast, the root growth inhibition ratio of alfalfa was only 21.539%, and the concentrations required for 50% growth reduction (IR50) for alfalfa roots was 1.757 g·L-1, which was 1.27 times that of amaranth (Figure 4A, Table 4). (page 5)
  • According to the non-linear fit of seedling growth, the significant effect of CA on amaranth stems was markedly depicted at the concentration of 1.432 g·L-1 by an 50% growth reduction (IR50) compared to alfalfa (21.271%), and the concentrations required for 50% growth reduction (IR50) for alfalfa stems was 1.769 g·L-1, which was 1.24 times that of amaranth (Figure 4B, Table4). (page 5)

  1. About the Intro, the idea is also to comment on other methods. Also, I think that only based on a regression excluding the classification was a very weak point of the design of the paper.

Response: We sincerely accept your comments. Therefore, we used non-linear fit to analyze the relationship between cyanamide concentration and its toxicity to alfalfa and amaranth. The fitting function is a common tool used to analyze the correlation. Many scholars usually use the fitting function to analyze the correlation of pesticide action and calculate the lethal concentration (LC50) and this analysis does not involve classification. Regression analysis between density of S. canadensis and the total number of local weed species in 1×1 m quadrat from different experimental plots[1].

 [1]  Yu, Zhang, Xiao-Ling, Song, Sheng, Qiang. Biological control of Solidago canadensis increasing biodiversity in invaded habitats using a bioherbicide.